# Relation between Dietary Habits, Physical Activity, and Anthropometric and Vascular Parameters in Children Attending the Primary School in the Verona South District

**DOI:** 10.3390/nu11051070

**Published:** 2019-05-14

**Authors:** Alice Giontella, Sara Bonafini, Angela Tagetti, Irene Bresadola, Pietro Minuz, Rossella Gaudino, Paolo Cavarzere, Diego Alberto Ramaroli, Denise Marcon, Lorella Branz, Lara Nicolussi Principe, Franco Antoniazzi, Claudio Maffeis, Cristiano Fava

**Affiliations:** 1Department of Medicine, University of Verona, 37129 Verona, Italy; alice.giontella@gmail.com (A.G.); angela.tagetti@libero.it (A.T.); pietro.minuz@univr.it (P.M.); denise.m@hotmail.it (D.M.); lorellabranz@yahoo.com (L.B.); lara.nicolussiprincipe@studenti.univr.it (L.N.P.); cristiano.fava@univr.it (C.F.); 2Department of Surgery, Dentistry, Paediatrics and Gynaecology, University of Verona, 37129 Verona, Italy; ire.bre@hotmail.it (I.B.); rossella.gaudino@univr.it (R.G.); paolo.cavarzere@ospedaleuniverona.it (P.C.); diegoalberto.ramaroli@univr.it (D.A.R.); franco.antoniazzi@univr.it (F.A.); claudio.maffeis@univr.it (C.M.)

**Keywords:** children, diet, physical activity, cardiovascular risk factors, obesity, hypertension, blood pressure, pulse wave velocity

## Abstract

The aim of this school-based study was to identify the possible association between diet and physical activity, as well as the anthropometric, vascular, and gluco-lipid parameters. We administered two validated questionnaires for diet and physical activity (Food Frequency questionnaire (FFQ), Children-Physical Activity Questionnaire (PAQ-C)) to children at four primary schools in Verona South (Verona, Italy). Specific food intake, dietary pattern, and physical activity level expressed in Metabolic Equivalent of Task (MET) and PAQ-C score were inserted in multivariate linear regression models to assess the association with anthropometric, hemodynamic, and gluco-lipid measures. Out of 309 children included in the study, 300 (age: 8.6 ± 0.7 years, male: 50%; Obese (OB): 13.6%; High blood pressure (HBP): 21.6%) compiled to the FFQ. From this, two dietary patterns were identified: “healthy” and “unhealthy”. Direct associations were found between (i) “fast food” intake, Pulse Wave Velocity (PWV), and (ii) animal-derived fat and capillary cholesterol, while inverse associations were found between vegetable, fruit, and nut intake and capillary glucose. The high prevalence of OB and HBP and the significant correlations between some categories of food and metabolic and vascular parameters suggest the importance of life-style modification politics at an early age to prevent the onset of overt cardiovascular risk factors in childhood.

## 1. Introduction

Since childhood, the presence of cardiometabolic risk factors such as obesity, hypertension, high levels of cholesterol, and triglycerides [1] may lead to the development of an atherosclerotic fatty streak in the intima of arteries [2,3]. Childhood obesity—considered by the “World Health Organization” (WHO) to be one of the most serious problem of 21^st^ century—is a well-known cause of noncommunicable disease in adults [4]. High blood pressure (HBP) in children is responsible for haemodynamic changes, including altered artery elasticity [5]. At a young age, the process is considered reversible [2] and can be prevented or minimized by implementing a healthy lifestyle [3]. Dietary habits and physical activity represent key points in the prevention of cardiovascular risk factors [6]. Diet is considered one of the major modifiable determinants of chronic diseases, with more and more scientific evidences supporting the fact that a “healthy” diet pattern could protect [5] a diet consisting of a daily intake of fruits and vegetables combined with a low consumption of salt, sugar, and saturated fat, in addition to industrially-produced-trans-fatty acids; this is still considered as a “high-quality” diet [5]. It is associated with a better cardiometabolic profile both in adults and in children [1,5]. Many studies have investigated the association between food intake and cardiovascular risk factors. With globalization and urbanization, children are exposed to ultra-processed, energy-dense, and nutrient-poor foods, which are cheap and readily available [7]. The intake of this kind of food increases the risk for young people to develop obesity in childhood and in adulthood [8]. To the contrary, “healthy diets” such as a Mediterranean-like dietary pattern rich in vegetables, fruit, fish, nuts, and olive oil, have been associated with not only to a reduction of blood pressure, improvement in lipid profile, endothelial function, and vascular inflammation in adults [9] but also to a reduction of inflammation already present in the younger population [10]. 

Physical activity influences body composition in terms of the amount of fat and muscle [7]. Because sedentary lifestyles are a rapidly increasing problem in both developed and developing countries [7], an appropriate level of physical activity provides health beneficial for musculoskeletal tissue, cardiovascular system, neuromuscular awareness, body weight control, and psychological benefits [11]. WHO reports that more than 80% of adolescents do not acheive the recommended level of daily physical activity [7]. WHO guidelines recommend that for at least one hour a day children engage in moderate-to vigorous physical activity, including games, sports, transportation, chores, recreation, physical education, or planned exercise, in the context of family, school, and community activities [12]. 

This study is a part of an observational school-based study set up with the aim to assess the relationship between food, physical activity, and the main cardiovascular risk factors in children attending their third and fourth class of primary school in the Verona South district. In particular, we aimed at assessing the prevalence of obesity (OB) and HBP in a defined age group and the possible associations between (i) dietary pattern and physical activity level, as well as (ii) anthropometric, gluco-lipid, and hemodynamic measures. This study could be useful to clarify how many children carry either overt or potential CV risk factors and additionally clarify how strong the association is between these factors and lifestyle.

## 2. Materials and Methods

### 2.1. Study Design

Children were recruited from the third and fourth classes of four primary schools in the Verona south district. The choice of the sample was determined by the age range (7–10), which was considered well suited for the aims of the study, as mostly prepuberal children were willing to participate. 

The Verona South district was chosen because of the headmasters’ willingness to allow us to perform the investigation. Subjects who refused to participate in the study or did not have their parents’ consensus were excluded. The study was conducted according to a cross-sectional observational design and was approved by the Ethical Committee of Verona and Rovigo (CESC) (*n* = 375). Written inform consent was signed by all children’s parents. 

In their school gym, children were evaluated in the morning starting at 8:30 a.m. Anthropometric measurements were collected with children wearing light clothes without shoes. Weight and height were measured with a calibrated balance and stadiometer; body mass index (BMI, kg/m2) was calculated. Children were classified as overweight (the percentile BMI for their age was above the 85th) or obese (the percentile BMI for their age was above the 95th) using the WHO child growth standard [13,14]. Waist-height ratio was measured and values were transformed in z-score and percentile [15,16]. Brachial blood pressure (BP) was the average of the three measurements in the supine positions, using a children-validated semiautomatic oscillometric device (Omron 750 IT) [17]; the measurments were expressed as a Z-Score and percentiles were indicated by the guidelines [18,19]. Subjects whose BP was between the 90th and 95th percentile were considered to have normal-high BP, while those with a BP equal or above the 95th percentile were defined as having a high BP. 

Besides brachial BP, even central aortic pressure waveform (cSBP, mmHg) and carotid-femoral pulse wave velocity (PWV, m/s) were derived using the SphygmoCor XCEL device. The cuff pulsations were recorded at the brachial artery, then a general transfer function was applied to calculate aortic waveform [20] using a cuff around the femoral artery that captures the femoral waveform and a tonometer that captures the carotid waveform. The velocity is computed by dividing the distance between the carotid and femoral arteries, using the pulse transit time. Z-score and percentiles were computed for cSBP and PWV [21,22]. Capillary cholesterol, triglycerides, and glucose were measured from fingerpick blood drops with two point-of-care testing (POCT) instruments (for cholesterol and triglycerides: HPS Multicare-in, Biochemical System International, Arezzo, Italy; for glucose: and Nova Biomedical, Waltham, MA, USA) [23,24], while children were fasting for at least four hours. Two questionnaires, both previously validated for children, were administered: “Food Frequency Questionnaire” (FFQ) [25] and a Physical Activity Questionnaire for Older Children (PAQ-C) [26,27]. Questionnaires were explained to the children and their parents on a previous informative day, then compiled at home along with parents and revised at the evaluation day with each child by a dedicated dietician.

### 2.2. Food Frequency Questionnaire (FFQ)

Children indicated their usual consumption of 61 items on the FFQ, using a 5-point scale (never; 1–2 times a month; 1–3 times weekly; 4–5 times weekly; one a day; more than once daily). Association of diet to diseases needed to be determined through different approaches because diet is a complex exposure variable [28]. Thus, we investigated a single FFQ intake (eggs, oil, seed oil, and nuts), main food groups intake (cereals and tubers, dairy products, legumes, fish, vegetables, fresh and dried fruit, meat, FFQ category of fast food, sweets, animal fat-derived condiments, junk food) and dietary patterns in relation to other collected variables. The patterns were extrapolated using the exploratory Principal Components Analysis (PCA), which represents one of the most used tools to derive behavioral patterns [29]. Dietary patterns could be representative of the intake of the usual combinations of individual food or groups of foods [29,30] and could provide more information regarding the association diet-disease since it reflects individual dietary behaviors [31].

### 2.3. Physical Activity

PAQ-C is a seven-day recall composed of nine statements about the frequency of physical activities at school, at home, and during leisure time. A score from 1 to 5 is assigned to each item and a mean total score of physical activity, the PAQ-C score, is then computed [32]. PAQ-C represents a valid and reliable method to assess general levels of physical activities but does not provide specific information about frequency, time, and intensity. For this reason, the first item of PAQ-C has been integrated with a semiquantitative question asked to define one’s physical activity level in terms of Metabolic Equivalent of Task (MET), which is defined as the energetic cost of sitting quietly [33]. A total MET-minutes/week has been obtained, as described in Appendix A. The threshold indicated in the “Guidelines for data processing and analysis of the IPAQ” of 600 and 3000 MET-minutes/week has been used, respectively, to categorize subjects in “low”, “medium”, or “high” adherence to moderate-vigorous physical activity [32].

### 2.4. Statistics

Data are expressed as mean ± standard deviation (SD) for continuous variables or percentages for categorical ones. The level of the p-value was set at 0.05. The Spearman correlation coefficient was used to quantify linear relationship between variables. The Student’s T-test was used to compare variables among two groups, while the ANOVA with Tukey post-hoc test was used for several groups. The relationship between categorical data was tested using the chi-square test. A power calculation based on an estimated prevalence of 9% obesity, as previously reported in Italian children [34], indicates that with our actual sample size of 300 children, we have 80% power to detect the true prevalence of obesity with a precision of 3.24% (that is a 95% CI of the estimate, between 5.76% and 12.24%). PCA was set as described in Appendix A. Adequacy of the sample to perform PCA were tested with Kaiser-Meyer-Olkin (KMO) test. Multivariate linear regression models were performed in order to test if diet pattern and/or physical activity remained associated with hemodynamic and metabolic variables, with results that were correlated by univariate analysis after adjustment. Age, sex, ethnicity, BMI, quartiles of kilocalorie intake, and quartiles of PAQ-C scores were used as covariates in the models. Statistics were performed with SPSS (IBM Corp. Released 2015. IBM SPSS Statistics for Windows, Version 23.0. Armonk, NY: IBM Corp, USA) and R (R Core Team (2014). R: A language and environment for statistical computing. R Foundation for Statistical Computing, Vienna, Austria). Graphs have been created with GraphPad Prism version 7.00 for Windows (GraphPad Software, La Jolla, San Diego, CA, USA).

## 3. Results

### 3.1. Characteristics of the Population

A total of 309 out of 413 children chose to participate in the study (participation rate 74.8%). Out of the 309 children recruited in the study (97.1% of population), 300 children (Age: 8.6 ± 0.7 years, male: 50%) filled out the questionnaires correctly. Baseline characteristics are shown in Table 1.

The prevalence of overweight and obesity were 21.3% and 13%, respectively, while the prevalence of normal-high BP and HBP were 17.6% and 21.7% (Figure 1), respectively. Among obese children, the prevalence of HBP was 30.8%, whereas among overweight children, the prevalence of HBP was 20.3%. Moreover, 150 (50%) and 121 children (41.3%) were found to have cSBP higher than either the 90th or 95th percentile for height, respectively. Among the 55 (18.3%) children classified as HBP by brachial SBP, 45 (81.8%) and 41 (78.8%) present cSBP over either the 90th or 95th percentile for height, respectively. We found higher consumption of vegetables in children categorized as normal weight group, compared to the overweight and obese group (Appendix A.), as well as in normotensive children compared to children with normal-high or HBP (Appendix A.).

### 3.2. FFQ

#### 3.2.1. Single FFQ Category

In univariate analysis, some significant correlations between the FFQ categories and anthropometric, haemodynamic, and gluco-lipid parameters were found (Table 2). In particular, we underlined the direct association between “fast food” and BMI, PWV, brachial, and central BP, capillary triglycerides. Meat intake was associated with a higher PWV and brachial DBP, whereas vegetables and fruit (fresh and nuts) intake were associated with a lower level of capillary glucose. Nut intake confirms the inverse correlation with capillary glucose that was already evident in the fruit group (r_s_ = -0.162, *p* < 0.05). Among condiments, those derived from animal fat (butter and lard) correlated positively with the level of capillary cholesterol, while among vegetable fats, seed oil intake is associated to a higher value of central systolic blood pressure (r_s_ = 0.117, *p* < 0.05). 

#### 3.2.2. Dietary Pattern

A KMO of 0.802 attested to the adequacy of data to perform factor analysis. PCA identifies two main patterns. The first principal component was represented by high factor loadings for fish, legumes, vegetables, fresh and dried fruit, and dairy products intake. This pattern was considered a “healthy” diet. Moreover, the second principal component, characterized by high factor loadings for cereals and tubers, sweets, fast food, meat, and eggs intake was considered a “unhealthy” pattern (Appendix A). Individual scores were used to correlate with specific parameters. (Appendix A). The “healthy” pattern correlated with lower level of capillary glucose (r_s_ = −0.191, *p* < 0.01), while the “unhealthy” pattern directly correlates with brachial DBP (r_s_ = 0.130, *p* < 0.05) (Appendix A).

### 3.3. Physical Activity

PAQ-C data were available for 286 (95.3%) children that completed the questionnaire correctly. Then, the PAQ-C score (1.8 ± 0.5) and total MET-min/week (3278.9 ± 3953.4) were computed (Table 1). The results found that 42 (14.0%) children spent less than 600 MET-min/week in moderate-vigorous activities. Further, 136 (50%) and 96 (35.3%), respectively, spent from 600 to 3000 and more than 3000 MET-min/week. MET min-week spent in moderate-vigorous activity was inversely correlated with central SBP, brachial DBP, and triglycerides. No significant correlation was found with the PAQ-C score (Appendix A). With regard to categories that asked about physical activity measured as MET, central SBP was significantly higher in the “low” category versus “medium” and versus “high” category (Appendix A) (Figure 2).

### 3.4. Multivariate Models

Linear regression models confirmed that even after adjustment for age, sex, ethnicity, and BMI, quartile of kcal intake and quartile of PAQ-C score had significant associations between fast food intake and PWV (β = 0.337; 95% CI: 0.140/0.534; *p* < 0.01). In addition, vegetable and fruit intake remained significantly associated to a decreased capillary glucose level [(β = −0.173; 95% CI: −3.251/−0.207; *p* < 0.05); (β = −2.530; 95% CI: −4.274/−0.785; *p* < 0.01)]. Nut intake was inversely associated to capillary glucose (β = −2.500; 95% CI: −0.011/−1.255; *p* < 0.05). Higher animal fat-derived condiment intake remained independently associated to a higher level of capillary cholesterol (β = 11.009; 95% CI: 1.349/20.669; *p* < 0.05) (Figure 3a–e) (Appendix A). The “healthy” pattern remained independently associated with capillary glucose levels after adjusting for cofounder variables (β = −0.016; 95% CI: −0.027/−0.005; *p* < 0.01), which is contrary to the association between the “unhealthy” pattern and brachial DBP (β = 0.911; 95% CI: −0.150/1.972; *p* = n.s.) (Appendix A)) (Figure 4a–b).

## 4. Discussion

Perhaps the most alarming result of the present study is the high prevalence of overweight-obesity and high blood pressure in children attending their third and fourth years of primary school in the Verona South district. More than 30% of children are overweight (21.3%) or obese (13.0%), which is in line with data reported in 2016 by the Italian National Surveillance System “Okkio alla salute” related to the prevalence of overweight and obese children (8–9 years) of 21.3% and 9.3%, respectively [34]. Further, in 2016 WHO estimated that 17% of children and adolescents (5–19 years) worldwide were overweight [7]. 

Our sample results show a high prevalence of normal high-BP and HBP (respectively 17.6% and 21.7%), with a further increased frequency in obese children. However, data about BP should be considered cautiously because despite having the average of three measurements, using a device validated in children, the measures were recorded in a single occasion, with children not in seated position, and thus cannot be considered diagnostic for hypertension. Moreover, in a subgroup of 25 children with BP ≥ 95th percentile, BP was measured in a follow-up visit in standard conditions, as requested in clinical guidelines, to confirm the presence of hypertension [19]. Only one child resulted to maintain a brachial BP higher than the 95th percentile for SBP or DBP, whereas two children were reclassified as normal-high BP (<90th percentile of brachial SBP or DBP <95th) and seventeen children had normal BP. It is important to note that of the children with brachial SBP higher than the 95th percentile, they also had a cSBP higher the 95th percentile for height, at least suggesting that these values were not driven by an abnormal amplification of the pressure wave, which is common in young subjects [35]. 

The prevalence of hypertension in the adult population of the United States and Europe has been estimated at nearly 15–30%, while the prevalence in children is 2–4%. However, it is common opinion that hypertension is under-diagnosed in clinical practice [36]. Moreover, the estimated percentage of school-children in Italy with high BP is reported to be 2–4% [37,38]. Many children today grow up in an obesogenic environment that encourages weight gain and obesity, caused by the changes in dietary habits and food availability and from the decline of physical activity [7]. Promoting the intake of healthy food and reducing the exposure of children to unhealthy dietary patterns is one of the programs declared by World Health Organization (WHO) to deal with childhood obesity [7]. From our data, it emerged that both “at risk” groups of overweight/obese children and those with normal-high/high BP values (percentile brachial SBP or brachial DBP >90°) eat less quantity of vegetables with respect to the normal weight or normotensive group at their same age. Many global health programs endorse an increased consumption of plant-based foods because of their inverse association with several chronic disease [39]. 

Data from the multi-center International Study of Asthma and Allergies in Childhood (ISAAC), which included 77243 children, demonstrated an inverse association between BMI and the consumption of vegetables [40]. Apart from the above-mentioned difference in vegetables intake, from our results, no clear association emerged between any type of food intake or grade of physical activity and markers of obesity (BMI and waist/height ratio). The relationship between BMI and fast food consumption has been widely reported. In a previous analysis, Braithwaith et al., within the aforementioned ISSAC, reported that more frequent fast food consumption was associated to a BMI of 0.14 kg/m2, which is higher than the infrequent consumption in a sample of 72900 children aged 6–7 [41]. Other studies reported the beneficial effect that physical activity has on weight [42,43,44], when physical activities was assessed using objective method like accelerometers. In a systematic review by Janssen et al., they included most of the studies with self- or parental-assessed physical activity, wherein the association between physical activity and weight was weak and associated to a not-significant risk [45] further underlining that data from questionnaire. 

Conversely, we found associations between the intake of some food groups and haemodynamic and gluco-lipid variables. In particular, fast food intake resulted to be associated to an increased PWV. This correlation, although weak, remained significant after adjusting for age, sex, ethnicity, BMI, quartiles of daily kilocalories intake, and quartile of PAQ-C score. Even if to our knowledge there is no study that directly correlates fast-food intake and PWV, it has already been reported that PWV is inversely associated to healthy lifestyle and Mediterranean Diet [46,47]. “The Cardiovascular Risk in Young Finns Study” collected lifestyle data from childhood and had a 27-year follow-up. They reported that vegetable intake in childhood is an independent predictor of PWV in adulthood [46]. 

In the EVIDENT cohort study, a higher adherence to the Mediterranean diet—expressed as EVIDENT, a diet index derived from the FFQ—was associated to lower values of PWV [47]. In another sample including 77 12-year-old children, adherence to the Mediterranean diet was estimated by the KIDMED index and had negative results that correlated with the Augmentation Index, independent of obesity [48]. 

As for gluco-lipid measurements, we found that a higher intake of animal-derived fat that are rich in cholesterol was associated with a higher level of capillary cholesterol consistent with the fact that cholesterol intake reflects cholesterol blood concentration [49]. We found also that glucose level was lower in children consuming most frequently vegetables and fruit (fresh and nuts). In a meta-analysis, Wang PY et al. investigated the influence of vegetable and fruit intake on the risk of type-2 diabetes. It emerged that several studies reported the beneficial role of these food groups [50]. Fruit and vegetables are rich sources of fiber, flavonoids, and anti-oxidant compounds (carotenoids, vitamin C and E), folate, and potassium, which could explain the protective effects of fruit and vegetables on type 2 diabetes [50]. 

Among fruit we found that especially nuts are associated to lower glucose level. Nuts have an increased interest in research for its association to healthy outcomes [51]. Nuts are mainly composed of polyunsaturated fatty acids (PUFA) in addition to complex carbohydrate and fiber, tocopherols, minerals, phytosterols, and polyphenols [52]. Its consumption, in moderate quantity, is considered a part of the Mediterranean Diet pattern and recommended to an all-world population [51]. In a previous study conducted in a sample of obese children, we found an inverse correlation between omega-6 fatty acids (FA), whose nuts are particularly rich, and several parameters of the metabolic syndrome including glucose, suggesting that this kind of fruit can be particularly healthful in children [53]. When nut intake is high, randomized control trials (RCT), a large prospective study and meta-analysis, showed improved cardiovascular disease (CVD) outcomes in adults [49]. Further, the PREDIMED trial study showed the beneficial effects of nut intake, as well as olive oil, on the occurrence of chronic diseases including cardiovascular disease and type 2 diabetes [54]. In other trials including patients affected by type-2 diabetes, a higher nut intake was associated with an improved blood glucose control [52]. Even if there is coherence with literature, a limitation of these evidences is collected from capillary levels of cholesterol, triglycerides, and glucose and children not properly fasting. 

Previous studies have shown that dietary patterns could be more strongly associated to executive functions than a single food intake [55]. This is due to the fact that FFQ has some intrinsic bias, such as the missing information about portion size that does not allow for a nutrient quantification. Through principal component analysis, we identified two patterns, defined as “healthy” and “unhealthy”. These patterns are slightly similar to the patterns identified in literature as the “Mediterranean” and the “Western” pattern [56]. The “healthy” pattern, composed of vegetables, fruits, legumes, and dairy products, reflects the combination of food characteristics of the Mediterranean Diet that is rich in minerals, vitamins, polyphenols, fibers, polyunsaturated fatty acids (PUFAs), and monounsaturated fatty acids (MUFAs) [57]. These nutrients are mainly considered protective against CVD through the modulation of blood pressure, lipid profile, body weight, and fasting blood glucose [53,58,59]. There is strong evidence that the promotion of the Mediterranean Diet along with physical activity results in healthier effects [60]. In our sample, the higher adherence to the “healthy” pattern was associated with lower glucose levels, whereas children classified as “more active” had lower values of central systolic BP and brachial diastolic BP. This association did not remain significant after adjustment for cofounder variables, in which only BMI remained significant, suggesting that BMI could mediate this deleterious hemodynamic effect. 

Physical activity contributes to a healthy lifestyle and it is a well-known fact that regular exercise is associated with minor incidence of future cardiovascular disease and mortality [61], which also contributes to epigenetic modifications in the regulation of CVD-associated genes [62]. Gerage et al. found a beneficial association of physical activity on central BP in a population of hypertensive adults [63], whereas in another study central, SBP was determined by body weight rather than exercise capacity in a population of children and adolescents [64]. We also found an inverse correlation between the total amount of MET min/wk spent in moderate physical activity and level of capillary triglycerides. Within the longitudinal “The Cardiovascular Risk in Young Finns Study”, Raitakari et al. found a lower triglyceride level among children and adolescents who spent more time in physical activity. However, like in our case, the association did not persist after adjusting for cofounders [65]. 

We did not find any significant association using the PAQ-C score. Obtaining accurate measures of physical activity is challenging, particularly in children [66]. The PAQ-C questionnaire is a convenient and cost-effective method [66] but has some limitations in terms of accuracy because children tend to overreport [66]. Moreover, there is an ongoing debate on the quantification of physical activity across different populations. The standardization of operating procedures aimed at assessing more objective physical activity like the ALPHA-fitness test battery, already adopted in adolescents, could allow a more direct comparison between children and adolescents in different countries [67,68]. The combined association of diet and physical activity was associated as expected to better cardiometabolic values. 

## 5. Conclusions

A strength of this work is that we investigated the lifestyle behavior in relation to cardiovascular risk factors in school-age children, highlighting the prevalence of obesity and high-blood pressure in this age group. This could be useful to detect children that could be considered at a higher risk of these things at earlier ages. Another strength is the measure of both central and brachial blood pressure the former being considered a better predictor of cardiovascular events than brachial blood pressure [64]. With regard to limitations, the cross-sectional design of our study represents a limitation for the emerged associations that cannot be considered causal, even if the biological plausibility of their link is high. Even the sample size is relatively limited and along with the choice of the sample, considerable “of convenience”, can imply a problem of generalizability of our results. Anyhow, as stated before, our data of prevalence especially for overweight and obesity are in line with other Italian surveys, and the sample size was sufficient to detect statistically significant correlations with a correlation coefficient as low as 0.13. There are also intrinsic limitations of the method of assessment of dietary and physical activity data using questionnaires, such as misreporting, usual food, and/or activities that are not reported in the list of questions. In conclusion, our results launch an alarm for the high prevalence of obesity in pre-adolescent children in Verona and also strengthen the importance that diet and physical activity have on vascular and metabolic function starting in childhood. Our data also supports the need to improve diet and physical activity programs as claimed from the World Health Organization (WHO).

## Figures and Tables

**Figure 1 nutrients-11-01070-f001:**
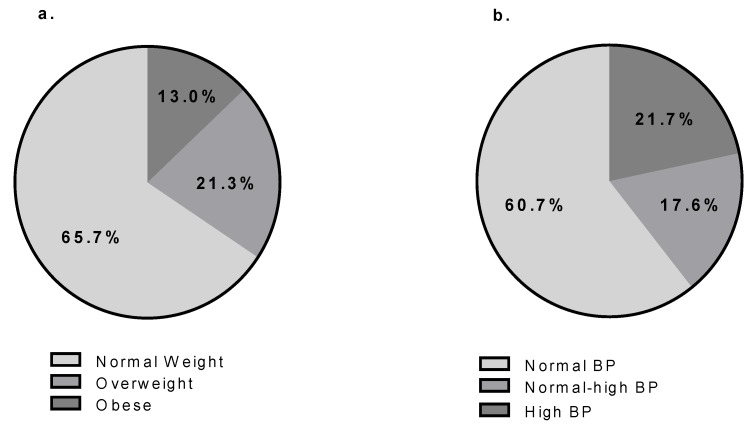
Prevalence of overweight/obese (**a**) and normal high/high blood pressure (BP) (**b**). Legend: “Normal weight”: percentile body mass index (BMI)—age < 85^th^; “Overweight”: 85^th^ < P percentile BMI—age < 95^th^; “Obese”: percentile BMI—age > 95^th^; “Normal blood pressure (BP)”: percentile SBP and DBP < 90^th^; “Normal-high blood pressure (BP)”: 90^th^ < percentile SBP or DBP < 95^th^; “High blood pressure (BP)”: percentile SBP or DBP > 95^th^.

**Figure 2 nutrients-11-01070-f002:**
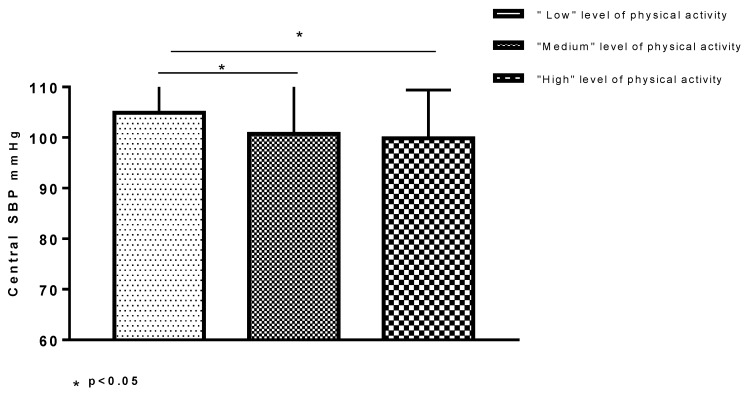
Difference of central blood pressure (cSBP, mmHg) among levels of moderate-vigorous physical activity. Legend: “Low”: <600 MET-minutes/week; “Medium”: 600–3000 MET-minutes/week; “High”: >3000 MET-minutes/week. SBP: systolic blood pressure.

**Figure 3 nutrients-11-01070-f003:**
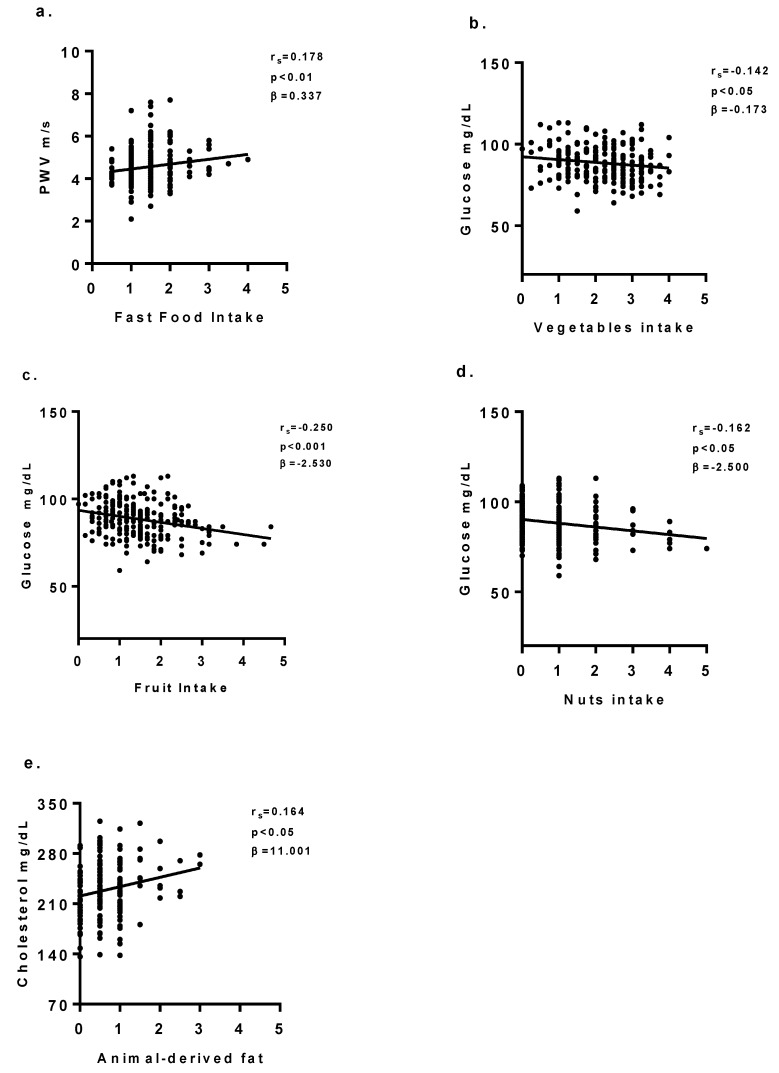
(**a–e**) Correlations between food intake and hemodynamic and gluco-lipid parameters. Correlation between: fast food intake and PWV (**a**); vegetables intake and glucose (**b**); fruit intake and glucose (**c**); nuts intake and glucose (**d**); animal-derived fat and cholesterol (**e**). PWV: Pulse Wave Velocity.

**Figure 4 nutrients-11-01070-f004:**
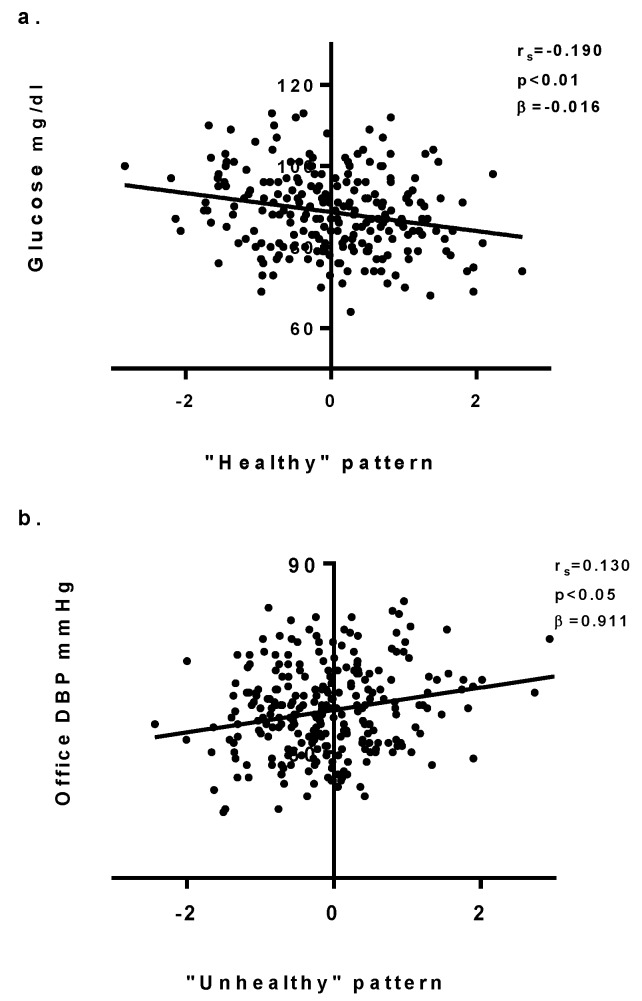
(**a–b**) Associations between “healthy pattern” and capillary glucose level (**a**) and between “unhealthy pattern” brachial diastolic blood pressure (**b**) DBP: diastolic blood pressure.

**Table 1 nutrients-11-01070-t001:** General characteristics of the population.

Characteristics	Male (*n* = 150)	Female (*n* = 150)	*p*-Value
	mean ± SD	mean ± SD	
Age; ys	8.7 ± 0.8	8.6 ± 0.7	n.s.
Caucasian ethnicity; *n* (%)	102 (68.0)	94 (62.7)	n.s.
Other ethnicities; *n* (%)	48 (32.0)	56 (37.3)	
BMI; kg/m2	18.1 ± 3.2	18.2 ± 3.6	n.s
BMI; percentile for age	63.9 ± 30.1	61.6 ± 31.6	n.s.
Normal weight; *n* (%)	95.0 (63.3)	102.0 (68.0)	
Overweight; *n* (%)	35.0 (20.3)	29.0 (19.3)	n.s.
Obese; *n* (%)	20.0 (13.4)	19.0 (12.7)	
Waist-height ratio	0.46 ± 0.8	0.46 ± 0.8	n.s.
Waist-height ratio; percentile	44.4 ± 31.7	48.0 ± 31.3	n.s.
Brachial SBP; mmHg	110.5 ± 9.5	110.2 ± 10.3	n.s.
Brachial DBP; mmHg	66.3 ± 7.4	67.2 ± 8.1	n.s.
Brachial SBP; percentile	75.9 ± 20.5	75.9 ± 20.5	n.s.
Brachial DBP; percentile	70.2 ± 19.2	71.6 ± 20.5	n.s.
Normal BP; *n* (%)	95 (63.3)	87 (58)	n.s.
High BP; *n* (%)	55 (36.7)	63 (42)	
Pulse Wave Velocity; m/s	4.6 ± 1.0	4.6 ± 0.8	n.s.
Pulse Wave Velocity; percentile for height	61.6 ± 40.2	58.4 ± 38.5	n.s.
cSBP; mmHg	100.2 ± 9.3	101.5 ± 10.7	n.s.
cSBP <90° percentile for height; *n* (%)	76.4 ± 27.2	78.1 ± 28.5	n.s.
cSBP >90° percentile for height; *n* (%)	68 (46.3)	78 (53.4)	n.s.
Capillary Triglycerides; mg/dl	178.4 ± 84.0	165.7 ± 62.4	*p* < 0.001
Capillary Cholesterol; mg/dl	241.5 ± 36.4	221.2 ± 38.5	n.s.
Capillary Glucose; mg/dl	92.3 ± 8.8	86.0 ± 10.4	*p* < 0.001
Energy intake kcal/die	2933.2 ± 932.0	3031.6 ± 1138.0	n.s.
PAQ-C Score	1.9 ± 0.4	1.8 ± 0.5	*p* < 0.05
Moderate-vigorous activity (MET-min/wk)	3825.2 ± 4230.7	2721.1 ± 3578.2	*p* < 0.05

BMI, Body Mass Index.

**Table 2 nutrients-11-01070-t002:** Correlations between food intake and anthropometric, hemodynamic, and gluco-lipidic parameters.

Characteristics	Fast Food	Cereals and Tubers	Vegetables	Fruit	Eggs	Meat	Dairy Product	Sweets	Legumes	Fish	Nuts	EVO oil	Animal-Derived Fat	Seed Oil
BMI, kg/m^2^	**0.129***	−0.080	−0.093	0.067	−0.028	0.042	0.102	−0.052	−0.020	−0.040	−0.049	0.076	0.057	0.07
Z-score BMI	**0.141***	−0.081	−0.090	0.044	−0.040	0.044	0.091	−0.039	−0.021	−0.030	−0.064	0.092	0.064	0.093
Waist-height ratio	0.062	−0.048	−0.105	0.025	−0.093	0.051	0.37	−0.045	−0.062	−0.085	−0.027	0.060	0.006	0.046
Z-score waist-height ratio	0.074	−0.036	−0.095	0.018	−0.106	0.044	0.031	−0.039	−0.083	−0.088	−0.041	0.057	0.007	0.034
Brachial SBP, mmHg	0.013	0.043	−0.052	0.106	−0.003	0.042	0.091	0.026	0.010	0.084	0.066	−0.091	0.064	**0.117***
Z-score Brachial SBP	−0.005	0.069	−0.051	0.087	−0.042	0.034	0.063	0.020	0.000	0.078	0.069	−0.260	0.025	0.076
Brachial DBP, mmHg	**0.124***	0.025	−0.101	0.006	0.019	**0.128***	0.027	0.048	−0.020	0.017	−0.057	−0.006	0.008	0.09
Z-score Brachial DBP	**0.118***	0.023	−0.107	−0.029	−0.010	**0.121***	0.009	0.061	−0.033	0.006	−0.057	−0.064	0.008	0.079
PWV, m/s	**0.178****	0.017	−0.021	**0.154****	0.035	0.110	−0.030	−0.022	0.037	0.013	0.040	−.056	0.048	0.111
Z-score PWV	**0.158****	0.002	−0.003	**0.130***	0.009	**0.126***	−0.052	−0.020	0.023	0.022	0.036	−0.075	0.011	0.090
cSBP, mmHg	**0.140***	0.021	0.011	0.102	−0.013	0.073	0.004	0.047	0.039	0.076	0.043	−0.075	−0.038	0.07
Z-score cSBP	**0.123***	0.026	0.009	0.082	−0.056	0.051	−0.021	0.032	0.026	0.053	0.050	−0.070	−0.028	0.07
C-Triglycerides mg/dl	**0.145***	−0.013	−0.062	−0.029	−0.072	−0.038	−0.024	−0.083	−0.077	−0.120	0.019	−0.039	0.023	−0.390
C-Cholesterol mg/dl	0.087	−0.062	0.029	−0.092	−0.029	−0.077	−0.129	−0.066	−0.011	0.074	0.096	−0.036	**0.164***	−0.015
C-Glucose mg/dl	−0.018	−0.088	**−0.142***	**−0.250****	−0.035	−0.066	−0.080	−0.088	−0.100	−0.110	**−0.16***	−0.024	0.071	−0.056

BMI: Body Mass Index; EVO oil: Extra Virgin Olive oil; SBP: systolic Blood Pressure; DBP: Diastolic Blood pressure; PWV: pulse wave velocity; c-: capillary. Significant Spearman correlations are expressed in bold (*= *p*-value<0.05; **= *p*-value<0.01)

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
