# Peer review of "Relation between Dietary Habits, Physical Activity, and Anthropometric and Vascular Parameters in Children Attending the Primary School in the Verona South District"

_nutrients, 2019, doi:10.3390/nu11051070_

Round 1

Reviewer 1 Report

Dear authors,

It is a pleasure to review an article like yours. Below I indicate the comments on it.

- Pag 1 line 11: Justify why diet is the most determining factor

Pag 2 linea 44: To unite both sentences, there is no grammatical cohesion when separated.

Indicate in method, how the sampling was done, that is, how the sample was chosen and why (ages, area, ...)

Was the FFQ validated beforehand? If so, indicate it, if it was not, indicate why, and explain better the choice of the questions and based on what they were done.

Add future lines: for example, include a better and greater study of body composition or fitness level through Alpha Fitness Battery ... (They have as an example the following article from the same publisher MDPI: https://www.mdpi.com/1660-4601/15/12/2632)

Check the bibliographical references, for example, the 16 does not seem to be in good format.

Author Response

We thank the reviewer for his/her suggestions and comments and have revised the manuscript accordingly.

Response to Reviewer 1’s comments

Point 1: Pag 1 line 11: Justify why diet is the most determining factor

Response 1: There is more and more evidence about how much diet has strong influences, both positively and negatively, on health and how dietary changes could determine whether or not an individual could develop chronic diseases. This is corroborated by the fact that the increase in chronic diseases over the past decades reflects the shift from a largely plant-based diets toward a high fat, highly processed and energy-dense diet.

Following this affirmation in the text, we wrote some examples on how different dietary patterns are associated to different cardiometabolic profiles.

Anyhow we acknowledge the fact that there also other important factors contributing to chronic disease and wrote in the manuscript “Diet is considered one of the major modifiable determinants of chronic diseases” (please see page 1-2 line 41-43)

Point 2: To unite both sentences, there is no grammatical cohesion when separated.

Response 2: There was a typing error, the sentence has been united (please see page 3, line 99)

Point 3: Indicate in method, how the sampling was done, that is, how the sample was chosen and why (ages, area, ...)

Response 3: The sample was chosen among the primary classes of Verona South, that is the Verona district in which our hospital is located. We have added this information (please see page 2 lines 74-78): “The choice of the sample was based on one side because the age range (7-10) was considered well suited for the aims of the study, with mostly prepuberal children, willing to participate. Verona South district was chosen because of the more easily availability by the headmaster to perform the investigation.

Possible limitations have been added to the discussion. (Please see page 5 lines 341-344).

Point 4: Was the FFQ validated beforehand? If so, indicate it, if it was not, indicate why, and explain better the choice of the questions and based on what they were done.”

Response 4: Both questionnaires, FFQ and Physical activity questionnaire were previously validated for older children. Food Frequency Questionnaire used is the semi-quantitative Youth/Adolescent FFQ (YAQ) developed by The Harvard T.H. Chan School of Public Health for the Nurses’ Health Study and validated by Rockett and colleagues (1997) for younger populations. We chose this because we thought was on one side comprehensive and consistent with our traditional consumed food, on the other hand relatively easy to fill in. We indicated in the text that the questionnaires are validated (please see pg 3 line 100,101 and references 22 for YAQ and 23-24 for IPAQ).

Point 5: Add future lines: for example, include a better and greater study of body composition or fitness level through Alpha Fitness Battery ... (They have as an example the following article from the same publisher MDPI: https://www.mdpi.com/1660-4601/15/12/2632)

Response 5:

Thank you for the suggestion. We are aware that PAQ-C has intrinsic limitation, even if it is widely used, validated and accompanied by guidelines that allow a classification in levels of physical activity. In the present study we had to mediate between the need to investigate one class per day and to collect sufficient information.

We acknowledge that the Alpha fitness battery would have allowed consistency of physical activity assessment across Europe. So, we added a part in the text explaining how can we improve this aspect, citing both the study the referee suggested and a study about the validation of Alpha fitness battery on young population. (please see page 5 lines 332-335)

Point 6: Check the bibliographical references, for example, the 16 does not seem to be in good format.

Response 6:

The reference has been fixed in the correct format (see reference 17).

Reviewer 2 Report

Paper was very well written. Although the study was not particularly novel, the investigators focused on an important population - overweight/obese children and incorporated lack of physical activity as part of the outcome variables. This made the paper of greater importance. 

Author Response

We thank the reviewer for his/her suggestions and comments and have revised the manuscript accordingly.

Response to Reviewer 2’s comments

Point 1: Paper was very well written. Although the study was not particularly novel, the investigators focused on an important population - overweight/obese children and incorporated lack of physical activity as part of the outcome variables. This made the paper of greater importance.

We thank the reviewer for the good comments about our manuscript and the importance of this study.

Reviewer 3 Report

The authors’ goal was to examine the cross-sectional relationships among anthropometric, dietary, activity and cardiometabolic variables in a sample of 309 children in 3rd -4th grade in South Verona, Italy. Understanding the relationships among cardiometabolic risk factors with children’s diet and activity are important to determining the impact of unhealthy dietary and activity patterns from a young age. While this line of research is worthwhile, there are a few concerns as noted below. Abstract: PWV is used without prior spelling out of the acronym. Introduction: Pg 2, Lines 49-52: The authors refer to a study that identified that consumption of foods as part of the Mediterranean diet was associated with reductions in cardiometabolic risk factors. However, the cited study, published 13 years ago, was conducted among older adults, aged 55 to 80 years. More recent studies (Carvalho, Ronca, Michels, et al., 2018) examine these relationships in adolescents, a population closer in age to participants in the current study. Methods Physical Activity (Sec 3.3): Pg 1, Line 189: The authors mention “central” SBP and “office” DBP, but there is no mention of what the terms “central” and “office” mean. Because the authors describe the measures being conducted at primary schools, these descriptors do not make sense. Results: The authors do not provide an effect size for their primary outcome. They mention in the discussion about the limitation of sample size (Discussion, Line 327). Considering the conclusions drawn, this is a concern.

Author Response

We thank the reviewer for his/her suggestions and comments and have revised the manuscript accordingly.

Point 1: PWV is used without prior spelling out of the acronym

Response 1: Thank you, we have now corrected it.

Point 2: Lines 49-52: The authors refer to a study that identified that consumption of foods as part of the Mediterranean diet was associated with reductions in cardiometabolic risk factors. However, the cited study, published 13 years ago, was conducted among older adults, aged 55 to 80 years. More recent studies (Carvalho, Ronca, Michels, et al., 2018) examine these relationships in adolescents, a population closer in age to participants in the current study

Response 2: We thanks the reviewer for the precious suggestion and have now added also the paper (Carvalho, Ronca, Michels, et al., 2018) he cited (please see page 2, lines 53,54 and new reference n. 8).

Point 3: Methods Physical Activity (Sec 3.3): Pg 1, Line 189: The authors mention “central” SBP and “office” DBP, but there is no mention of what the terms “central” and “office” mean. Because the authors describe the measures being conducted at primary schools, these descriptors do not make sense.

Response 3: The reviewer is right and we are sorry not to be enough clear about “central BP definition”. Actually, “central” SBP refers to the estimate of central aortic pressure waveform (cSBP) as described in the section 2.1 Study design. And, as suggested, it is better to define the other BP as “brachial BP” since it is anyhow an out-of-office measurement. We have now corrected these issues throughout the manuscript.

Point 4: Results: The authors do not provide an effect size for their primary outcome. They mention in the discussion about the limitation of sample size (Discussion, Line 327). Considering the conclusions drawn, this is a concern.

Response 4: The reviewer is right. We have now provided a power calculation for our study regarding the prevalence of obesity. As reported in the methods section “A power calculation based on an estimated prevalence of obesity of 9%, as previously reported in Italian children [31], indicates that with our actual sample size of 300 children, we have 80% power to detect the true prevalence of obesity with a precision of 3.24% (that is a 95%CI of the estimate is between 5.76% and 12.24%).” (please see page 3 lines 136-140 and reference n. 31).

We have also expanded the discussion about the limitation of the sample size. (please see page 5 lines 342-348)

Round 2

Reviewer 1 Report

Accept in present form

Reviewer 3 Report

The authors have made all requested changes to the manuscript.